# Insights into PCSK9-LDLR Regulation and Trafficking via the Differential Functions of MHC-I Proteins HFE and HLA-C

**DOI:** 10.3390/cells13100857

**Published:** 2024-05-17

**Authors:** Sepideh Mikaeeli, Ali Ben Djoudi Ouadda, Alexandra Evagelidis, Rachid Essalmani, Oscar Henrique Pereira Ramos, Carole Fruchart-Gaillard, Nabil G. Seidah

**Affiliations:** 1Laboratory of Biochemical Neuroendocrinology, Montreal Clinical Research Institute (IRCM), University of Montreal, Montreal, QC H2W 1R7, Canada; sepideh.mikaeeli@mail.mcgill.ca (S.M.); ali.ben.djoudi.ouadda@ircm.qc.ca (A.B.D.O.); alexandra.evagelidis@ircm.qc.ca (A.E.); rachid.essalmani@ircm.qc.ca (R.E.); 2Département Médicaments et Technologies pour la Santé (DMTS), Université Paris-Saclay, CEA, INRAE, SIMoS, 91191 Gif-sur-Yvette, France; oscar.pereira-ramos@cea.fr (O.H.P.R.); carole.fruchart@gmail.com (C.F.-G.)

**Keywords:** PCSK9, LDLR, HFE, HLA-C, MHC-I, lipid metabolism

## Abstract

PCSK9 is implicated in familial hypercholesterolemia via targeting the cell surface PCSK9-LDLR complex toward lysosomal degradation. The M2 repeat in the PCSK9’s C-terminal domain is essential for its extracellular function, potentially through its interaction with an unidentified “protein X”. The M2 repeat was recently shown to bind an R-x-E motif in MHC-class-I proteins (implicated in the immune system), like HLA-C, and causing their lysosomal degradation. These findings suggested a new role of PCSK9 in the immune system and that HLA-like proteins could be “protein X” candidates. However, the participation of each member of the MHC-I protein family in this process and their regulation of PCSK9’s function have yet to be determined. Herein, we compared the implication of MHC-I-like proteins such as HFE (involved in iron homeostasis) and HLA-C on the extracellular function of PCSK9. Our data revealed that the M2 domain regulates the intracellular sorting of the PCSK9-LDLR complex to lysosomes, and that HFE is a new target of PCSK9 that inhibits its activity on the LDLR, whereas HLA-C enhances its function. This work suggests the potential modulation of PCSK9’s functions through interactions of HFE and HLA-C.

## 1. Introduction

The proprotein convertase subtilisin/kexin type 9 (PCSK9), discovered in 2003 by Seidah et al. [1], is the ninth member of the proprotein convertase (PC) family and is primarily expressed in hepatocytes. PCSK9 is the third gene implicated in familial hypercholesterolemia (FH3) because of its ability to target the low-density lipoprotein receptor (LDLR) to lysosomes for degradation in a non-enzymatic fashion [2], thereby increasing the circulating levels of LDL-cholesterol (LDLc) [3,4]. Accordingly, a number of strategies were proposed to silence PCSK9 activity in circulation and/or in hepatocytes, such as inhibitory monoclonal antibodies, siRNA, and CRISPR editing, resulting in 50–60% reductions in LDLc and significantly decreased cardiovascular events [3,5,6,7].

Structurally, PCSK9 comprises five distinct domains, including a signal peptide, a prodomain, a catalytic domain that interacts with the EGF-A domain of LDLR, a hinge domain, and a C-terminal domain known as the Cys-His-rich domain (CHRD), composed of three repeat structures termed M1, M2, and M3 [8,9]. Although the CHRD does not affect the binding of PCSK9 to the LDLR, it is required for the extracellular activity of PCSK9 to induce LDLR degradation [3,10,11,12].

The degradation of the LDLR by PCSK9 proceeds via two distinct pathways, occurring extracellularly [10,11,12] or intracellularly [3,12]. In the extracellular pathway, secreted PCSK9 binds the EGF-A domain of the LDLR on the cell surface, and the PCSK9-LDLR complex is then internalized in heavy-chain clathrin-coated vesicles and sorted to endosomes/lysosomes for degradation [10,12]. Thus, in the presence of PCSK9, LDLR is no longer able to recycle back to the cell surface to uptake more LDLc, leading to reduced levels of LDLR at the plasma membrane, and consequently to increased levels of LDLc in circulation. In contrast, the intracellular pathway shunts the PCSK9-LDLR complex from the *trans*-Golgi network (TGN) to lysosomes directly via light-chain clathrin-coated vesicles [3], before PCSK9 secretion and LDLR surface localization. These two sorting pathways collectively contribute to the regulation of LDLR levels and ultimately impact LDLc homeostasis. Liver hepatocytes are the main source of circulating PCSK9, and the extracellular pathway is the primary route of PCSK9-LDLR degradation [3].

The mechanism by which the extracellular PCSK9-LDLR complex is sorted to lysosomal compartments is not fully understood. However, the CHRD domain appears to play a critical role in this process by facilitating the interaction between PCSK9 and an unidentified partner protein referred to as “protein X” [12,13]. This interaction is essential for directing the PCSK9-LDLR heterodimer to endosomes/lysosomes for degradation. So far, several secretory proteins have been proposed as potential candidates for “protein X”, including APLP2 and Sortilin. However, none of them could be validated. Interestingly, the cytosolic adenylyl cyclase-associated protein 1 (CAP1) was reported to bind the M1 and M3 domains of the CHRD of PCSK9 and somehow enhance its extracellular activity on the LDLR [14]. The rationale for this binding became clear after we showed that CAP1 is secreted, and then binds the M1, M3, and prodomains of PCSK9, allowing optimal exposure of the M2 domain and thereby enhancing its extracellular activity, but it is not crucial [13].

The M2 domain of PCSK9 was reported to be key for its extracellular activity on the LDLR, suggesting that it interacts with a hypothesized “protein X” [12,13]. Befittingly, a number of natural variants in the M2 domain including the gain-of-function (GOF) H553R and loss-of-function (LOF) Q554E led to higher and lower circulating LDLc levels, respectively [15]. Recently, it was suggested that the M2 domain of PCSK9 binds an R-x-E motif in some MHC-class-I proteins (e.g., HLA-A), sending them to lysosomal degradation [16]. Among the nine-membered family of human MHC-I proteins [17], two specific members, including leukocyte antigen C (HLA-C: implicated in the immune system) and homeostatic iron regulator protein (HFE: involved in iron signaling), have captured our interest because of their likely involvement in lipid metabolism [18,19]. Recently, the 3D structure of HLA-C and its interaction with PCSK9 was modeled [13]. This work confirmed the importance of the R-x-E motif (Arg_68_ and Glu_70_) of HLA-C for its interaction (with Glu_567_ and Arg_549_) in the M2 domain of PCSK9, respectively. Removal of the interacting Arg and Glu in PCSK9 or HLA-C led to a complete LOF in the activity of PCSK9 on the LDLR. This compelling observation proposed HLA-C (and/or another MHC-I member) as a potential “protein X” that is necessary for the extracellular function of PCSK9 on the LDLR [13].

In 2020, an interesting study conducted by Demetz et al. uncovered a novel role for HFE that extends beyond its established function in iron signaling. In this work, the authors demonstrated that siRNA silencing of HFE expression in HepG2 cells resulted in elevated levels of LDLR [19]. Notably, they observed that mice carrying the HFE C282Y mutation that abrogates β2-microglobulin binding, leading to HFE retention in the endoplasmic reticulum (ER), displayed higher LDLR levels compared to wild-type (WT) mice [19]. Furthermore, a meta-analysis conducted in 2009 revealed a significant association between the HFE C282Y variation and lower levels of LDLc (−15 mg/dL) [20]. While these findings have indeed uncovered a new and important role for HFE in lipid metabolism, the specific mechanism through which HFE is implicated in intracellular or extracellular LDLR regulation remains an open question.

In this work, our research focused on examining how HFE may regulate extracellular PCSK9 activity on the LDLR. We also confirmed the involvement of HLA-C as a potential “protein X” and compared the trafficking pathways of HFE and HLA-C as possible opposing regulators of PCSK9 in this process. Our study indicated that HLA-C and HFE exert opposite effects on PCSK9, possibly through two distinct regulatory pathways.

## 2. Materials and Methods

### 2.1. Generation of Constructs

Human complementary DNAs (cDNAs) encoding wild-type and mutant forms of LDLR, PCSK9, HFE, and HLA-C (HFE and HLA-C WT cDNA purchased from Genscript) were generated through site-directed mutagenesis. These cDNAs were incorporated into vectors such as pIRES2-EGFP or pcDNA3.1+/C-(K)-DYK for expression. Additionally, both negative and positive control constructs were included in the experimental setup. To distinguish and track the expressed proteins, various tags like V5 and FlagM2 were introduced to the constructs. Before further analyses, the sequence integrity of each mutant construct was rigorously confirmed using Sanger DNA sequencing. Point mutations or deletion mutants were generated through a 2-step polymerase chain reaction (PCR) technique as previously described and verified by DNA sequencing.

The following primers were used for mutagenesis:PCSK9-ΔM2FP-CTACCCCAGCCAGGTCTGGAATGCRP-TCCAGACCTGGCTGGGGTAGCAGGCAGPCSK9-H553RFP-CCACTGCCGCCAACAGGGCCRP-CTGTTGGCGGCAGTGGACACPCSK9-Q554EFP-CTGCCACGAACAGGGCCACRP-CCTGTTCGTGGCAGTGGACPCSK9-R549A-E567AFP-CATGGGGACCGCTGTCCACTGCCRP-GGCAGTGGACAGCGGTCCCCATGFP-GCAGCTCCCACTGGGCGGTGGAGGACCTTGGCRP-GCC AAG GTC CTC CAC CGC CCA GTG GGA GCT GCPCSK9-R549A-Q554E-E567AFP-CTGCCACGAACAGGGCCACRP-CCTGTTCGTGGCAGTGGACFP-CATGGGGACCGCTGTCCACTGCCRP-GGCAGTGGACAGCGGTCCCCATGFP-GCAGCTCCCACTGGGCGGTGGAGGACCTTGGCRP-GCC AAG GTC CTC CAC CGC CCA GTG GGA GCT GCLDLR-ΔCTFP-CTATGGACCGGTAAGCCTATCCCTAACRP-GATAGGCTTACCGGTCCATAGAAGGAAGACCCCCHFE-C282YFP-GAGCAGAGATATACGTACCAGGTGGAGCACCCAGGCCTGGRP-CCAGGCCTGGGTGCTCCACCTGGTACGTATATCTCTGCTCHFE-R67A-E69AFP-CATGAGAGTCGCGCTGTGGCGCCCCGAACTCCRP-GGAGTTCGGGGCGCCACAGCGCGACTCTCATGHLA-C-R68A-E70AFP-GCGACGCCGCGAGTCCGGCAGGGGCGCCGCGGGCGCCGTGRP-CACGGCGC CGCGGCGCCCCTGCCGGACTCGCGGCGTCGC

### 2.2. qPCR and Sequence of Primers

Quantitative RT-PCR was performed as published before [13]. In summary, a monolayer of cells grown on a 35 mm plate was lysed and homogenized using a QIAshredder spin column (Qiagen, Venlo, The Netherlands). Total RNA was isolated with an RNeasy mini kit (Qiagen). Synthesis of cDNA was performed as per manufacturer’s protocol using SuperscriptTM II RT (Invitrogen, Waltham, MA, USA) from 250 ng of total RNA. Quantitative PCR was performed with PowerUp™ SYBR™ Green Master Mix (Applied Biosystems™, Bedford, MA, USA) using the VIIA 7 Real-Time PCR system (Applied Biosystems™). Gene expression was normalized to that of the Tata-binding protein (TBP).

The following primers from Kruse et al. [21] were used for qPCR:TBP-FPCGAATATAATCCCAAGCGGTTTTBP-RPGTGGTTCGTGGCTCTCTTATCCPCSK9-FPATCCACGCTTCCTGCTGCPCSK9-RPCACGGTCACCTGCTCCTGHLA-A-FPCGACGCCGCGAGCCAGAHLA-A-RPGCGATGTAATCCTTGCCGTCGTAGHLA-B-FPGACGGCAAGGATTACATCGCCCTGAAHLA-B-RPCACGGGCCGCCTCCCACTHLA-C-FPGGAGACACAGAAGTACAAGCGHLA-C-RPCGTCGTAGGCGTACTGGTCATAHLA-E-FPCCTACGACGGCAAGGAHLA-E-RPCCCTTCTCCAGGTATTTGTGHLA-F-FPGGCAGAGGAATATGCAGAGGAGTTHLA-F-RPTCTGTGTCCTGGGTCTGTTHLA-G-FPTTGGGAAGAGGAGACACGGAACAHLA-G-RPAGGTCGCAGCCAATCATCCACHFE-FP(Origene #HP200390)HFE-RP(Origene #HP200390)β2M-FPCTGGGTTTCATCCATCCGACAβ2M -RPTTCACACGGCAGGCATACTCATC

### 2.3. Inhibition of Protein Expression by Small-Interfering RNAs (siRNAs)

siRNA analysis was performed using INTERFERin^®^ (PolyPlus, New York, NY, USA) transfection reagent according to the manufacturer’s instructions. The following siRNAs with a final concentration of 60 nM were used: CTL siGENOME non-Targeting siRNA Pool #2 (#D-001206-14-05), ON-TARGETplus Human CLTC (1213) siRNA-SMARTpool (#L-004001-01-0005), ON-TARGETplus Human CAV1 (857) siRNA -SMARTpool (#L-003467-00005), and siGENOME Human LDLR siRNA–SMARTpool (#M-011073-01-0005). All siRNAs were purchased from Dharmacon (Horizon Discovery, Cambridge, UK). Gene silencing efficiency was assessed by Western blotting.

### 2.4. Cell Culture and Transfection

Various cell lines were utilized: HEK293 (human-embryonic-kidney-derived epithelial cells), HepG2-naïve (human hepatocellular carcinoma) cells, the sub-clone CHO-K1 cell line from the original Chinese hamster overy cells (CHO), CRISPR HepG2 HLA-C^−/−^ cells (Ubigene, Inc., Guangzhou, China #YC-C001), and CRISPR HepG2 PCSK9^−/−^ cells [13]. These cells were cultured in specific growth media: Dulbecco’s Modified Eagle Medium (DMEM) or Eagle’s Minimum Essential Medium (EMEM) supplemented with 10% fetal bovine serum (FBS; GIBCO BRL). The cells were maintained at a temperature of 37 °C in an environment with 5% CO_2_ to simulate physiological conditions. Transfection was employed to introduce the desired genetic constructs (PCSK9, LDLR, HFE, HLA-C, and their variants) into the cells. Depending on the cell line, different transfection reagents were used: JetPEI (PolyPlus), FuGENE^®^HD (Promega, Madison, WI, USA), and jetPRIME (PolyPlus) transfection reagents for CHO-K1, HepG2, and HEK293 cells, respectively. Cells were allowed to express the introduced genes for 48 h post transfection. For HEK293 cells, a specialized protocol was followed: cells were coated with poly-L-lysine, and then seeded in large flasks (T175) to produce PCSK9-enriched media. jetPRIME transfection reagent was used for this process. After 48 h, the conditioned media containing the secreted protein were collected, measured by Elisa, and stored at a temperature of −80 °C for subsequent analysis. A similar production method was used for all experiments. For the media swap experiment, different cells were seeded in 12-well cell culture plates, and after 24 h, they were incubated with serum-free media overnight. Subsequently, cells were exposed to conditioned media produced from HEK293 cells overexpressing human PCSK9.

### 2.5. In-House ELISA Measurement of Human PCSK9 Levels in Media

The secreted concentrations of PCSK9 in the media were determined using an in-house luminescence-based human PCSK9 ELISA assay [13], which was conducted as follows: LumiNunc Maxisorp white assay plates were used and coated with 0.5 μg/well of anti-human PCSK9 antibody (hPCSK9-Ab). The coating was carried out by incubating the plates at 37 °C for 3 h and then at 4 °C overnight. After the coating, the plates were subjected to washing steps to remove any unbound components. The plates were then blocked using a blocking buffer composed of PBS (Phosphate-Buffered Saline), casein at 0.1% concentration, and Merthiolate at 0.01% concentration. Calibrators were prepared by creating serial dilutions of known concentrations of a standard PCSK9 solution. Samples, which contained secreted PCSK9 from the cell culture media, were prepared by diluting them at two different dilution ratios, 1:50 and 1:100, using a dilution buffer with BSA (Bovine Serum Albumin). The calibrators and samples were added to the coated and blocked plates and allowed to incubate for 30 min at a temperature of 46 °C. After the incubation, the plates were washed again to remove any unbound materials. Subsequently, a secondary antibody known as hPCSK9-Ab-HRP (Horseradish Peroxidase) was added to the plates. The plates were then incubated for 3 h at a temperature of 37 °C while shaking at 300 rpm. After the secondary antibody incubation, plates were washed once more. A substrate solution, specifically SuperSignal™ ELISA Femto Substrate from Pierce (ThermoFisher, Waltham, MA, USA), was added to each well of the plate. The generated chemiluminescence was quantitated using a Pherastar luminometer from BMG Labtech. The concentrations of the secreted PCSK9 in the samples were calculated and adjusted accordingly for each experimental construct, allowing for a comparative analysis of PCSK9 secretion across different conditions or treatments.

### 2.6. Western Blotting

Cultured cells underwent the following process for protein extraction and analysis: First, the cultured cells were washed to remove any residual media or contaminants. Then, a non-denaturing cell lysis buffer was used for protein extraction. The composition of the lysis buffer was as follows: 20 mM Tris-HCl (pH 8), 137 mM NaCl, 2 mM Na_2_EDTA, 1% NP-40 (Nonidet P-40), 10% glycerol, and 4% protease inhibitor cocktail (PIC) without EDTA. Then, a Lowry assay was employed to determine protein concentrations in the extracted samples. In the next step, the extracted proteins were separated by size using polyacrylamide gel electrophoresis (SDS-PAGE). Two types of gels were used: 6.5% and 8% tris-glycine gels. The separated proteins were then transferred from the gel onto PVDF (Polyvinylidene Fluoride) membranes and were incubated with specific primary antibodies that bind to the target proteins of interest. After the primary antibody incubation, secondary antibodies conjugated with Horseradish Peroxidase (HRP) were applied. The membranes were analyzed and quantified using a ChemiDoc imaging system from Biorad. For quantification of Western blot data, we normalized all samples to their corresponding internal control (tubulin) and then set the control (untreated condition) to one. The purpose of normalizing our controls was to mitigate variations between each blot/experiment, as the absolute intensity differed among blots due to factors such as variations in band intensity. The following antibodies were used in this work: α-tubulin (ProteinTech, Rosemont, IL, USA #11224-1-AP [1:10,000]), HFE (Santa Cruz, Dallas, USA #sc-514405 [1:100]), HLA-C (Santa Cruz #sc-166134 [1:500]), clathrin Heavy Chain (Abcam, Toronto, ON, Canada #ab21679 [1:1000]), caveolin-1(D46G3) (NEB-cell signaling, Ontario, Canada #3267T [1:1000]), hPCSK9 (in house [1:2000]), LDLR (R&D system #AF2148 [1:1000])), V5 (Invitrogen, Waltham, MA, USA #R960-25 [1:5000]), monoclonal ANTI-FLAG^®^ M2-Peroxidase (HRP) antibody produced in mouse (Sigma, MA, USA #A8592-1MG [1:10,000]), β2M (ThermoFisher, MA, USA #701250 [1:1000]), anti-mouse HRP (VWR, Radnor, PA, USA #CA95017-332L [1:10,000]), anti-rabbit HRP (VWR #CA95017-556L [1:10,000]), and anti-goat-HRP (#A5420 [1:10,000]).

### 2.7. Immunofluorescence Assay (IF)

For the IF experiment, CRISPR HepG2 PCSK9 KO cells were cultured, and their medium was replaced with a medium containing 0.3 ng/mL of human PCSK9 (hPCSK9). After the medium swap, cells were incubated for 24 h. After 48 h of incubation with PCSK9 followed by serum-free medium, these cells were washed twice with PBS (Phosphate-Buffered Saline) to remove any residual substances. Subsequently, they were fixed using 4% paraformaldehyde. To prevent nonspecific binding of antibodies, the fixed cells were blocked with a solution of PBS containing 2% BSA (Bovine Serum Albumin) for 1 h. Then, they were incubated with proper primary antibodies including LDLR (R&D system #AF2148 [1:200]), and EEA1 (Abcam #2900 [1:500]) at a temperature of 4 °C overnight. The next day, plates were washed with PBS to remove unbound primary antibodies and were then incubated with an appropriate fluorescent secondary antibody, including goat-Alexa 488 (Molecular probes, Oregan, USA #A-11078 [1:500]) and rabbit-Alexa 555 (Molecular probes #A-31572 [1:500]). To visualize cell nuclei, samples were stained with Hoechst dye at a concentration of 1 μg/mL. The coverslips containing the stained cells were mounted onto glass slides using Mowiol, a mounting medium. These prepared samples were then visualized using a confocal laser scanning microscope with a high-powered objective lens (Plan-Apochromat 63 × 1.4 oil) from Carl Zeiss, Baden-Württemberg, Germany. For quantification, three separate experiments were conducted. In each separate experiment, approximately 10 pictures per condition were captured using confocal microscopy. Within each picture, approximately 3–15 cells were analyzed (the mean intensity values were measured) for quantification.

### 2.8. PCSK9–LDLR (EGF-AB Peptide) Binding Assay

The CircuLex human PCSK9 functional assay kit (MBL MBL life science, Woburn, USA, Cat #CY8153) was used to measure the binding affinity of wild-type (WT) PCSK9 to LDLR. Media from HEK293 cells containing WT PCSK9 were incubated with HepG2 PCSK9 KO cells that transfected either with HFE or an empty vector. Then, samples were collected and serially diluted. These diluted samples were then used for the binding assay. LumiNunc Maxisorp white assay plates were coated with the recombinant LDLR EGF-AB domain. Serially diluted samples of PCSK9 were added to the coated plates containing the LDLR EGF-AB domain. For each concentration of PCSK9, the absorbance at 450 nm (OD) was measured using a SpectraMax i3 plate reader. The obtained OD values were corrected for nonspecific binding and normalized to the maximum absorbance value (OD/ODmax). A binding curve was generated for each PCSK9 variant using a 4-parameter logistic (4-PL) equation. The EC50 value, which represents the concentration of PCSK9 needed for half-maximal binding to the LDL receptor EGF-AB domain, was extracted from the binding curve.

### 2.9. Modeling of PCSK9/HFE Complex

GlobalRAngeMolecularMatching (GRAMM*, see https://gramm.compbio.ku.edu/, accessed on 11 March 2024) webserver was used for molecular docking between HFE complexed with β-2-microglobulin (PDB: 1A6Z; chains: A and B; assumed as a receptor) and PCSK9’s CHRD (PDB: 2P4E; assumed as a ligand). HFE residues R_67_ and E_69_ of the RVE motif (UNIPROT: Q30201; residues 45 and 47 in the crystallographic structure) were taken as interface constraints for filtering the 10 top models. The comparison of the structural models of the PCSK9/HLAC and PCSK9/HFE complexes was carried out using the PCSK9/HFE model described in this work and the PCSK9/HLAC model published in 2023 [13] using PyMOL.

### 2.10. Modeling of the Interaction between PCSK9’s N-Terminus with HLA-C and Other HLA Members

The ternary complex comprising PCSK9’s structured N-terminal peptide (uniprot: Q8NBP7; residues 31 to 59), the extracellular region of HLA-C’s α-chain (uniprot: P10321-1; residues 26 to 300), and β2-microglobulin (uniprot: P61769; residues 22 to 119) was modeled using Alphafold 2.3.1 in IDRIS HPC using NVIDIA V100 nodes (options: model_preset = multimer;use_gpu_relax;max_template_date = 2022-0101; num_multimer_predictions_per_model = 3). To verify if the best ranked model was compatible with previous observations of interactions between the α-chain of HLA-C and the CHRD of full-lengthPCSK-9 [13], a merged model was constructed by superposing both models using HLA-C/β2-microglobulin’s main chain as reference. Then, HLA-C/β2-microglobulin and PSCK9 peptide 31–59 of the previous model were removed. The PSCK9 peptide 31–59 (not interacting with the α-chain of HLA-C) was manually remodeled using Pymol (The PyMOL Molecular Graphics System, Version 2.5, Schrödinger, LLC, New York, NY, USA) and energy minimization using Amber ff14SB (200 steps steepest descent + 10 gc) implemented in Chimera [22].

Alternatively, the helical N-terminal peptide PCSK9 comprising residues 32–50 (PDB: 6MV5, complexed with anti-PCSK9 fab) was compared to the HLA-C α-1 domain using GRAMM [23] and defining the peptide residues E_34_, D_37_, and E_40_ as interface constraints. Before peptide–protein docking, the side chains of the HLA-C α-1 domain were relaxed using the fixbb application in the Rosetta package [24]. A similar methodology was used to predict the interaction of PCSK9’s N-terminal peptide with other HLA members including HLA-A, HLA-B, HLA-F, HLA-G, and HFE.

## 3. Results

### 3.1. In Search for “Protein X”

Analyses of PCSK9’s reported structures indicated that its M2 domain may interact with an unidentified protein, referred to as “protein X”, which is essential for PCSK9’s extracellular function. Prior studies proposed “protein X” as a transmembrane protein with a cytosolic tail capable of internalizing the PCSK9-LDLR and PCSK9-LRP1 complexes [3,25,26] in the absence of the cytosolic tail (CT) of the LDLR (LDLR-ΔCT). Herein, we investigated the role of the M2 domain–“protein X” interaction on PCSK9-LDLR trafficking by transfecting HEK293 cells with cDNAs encoding PCSK9-WT-V5, PCSK9-ΔM2-V5, and a control empty pIRES-EGFP vector to produce PCSK9-enriched media [13]. HepG2 PCSK9 KO CRISPR cells were then incubated with these media (~300 ng/mL of each PCSK9 construct, estimated by ELISA) and their effects on LDLR levels were analyzed using immunofluorescence microscopy under non-permeabilized (cell surface LDLR) and permeabilized (total LDLR) cellular conditions. The results showed that WT PCSK9 significantly reduced LDLR levels both at the cell surface and intracellularly, while PCSK9-ΔM2 had a minimal impact on intracellular LDLR, but greatly reduced cell surface LDLR (Figure 1A). These data suggest that the M2 domain is not essential for the initial cell surface binding of PCSK9 to the LDLR or cell surface internalization into early endosomes, as reported previously [12], but is rather critical for the ensuing endosomal trafficking of the PCSK9-LDLR complex to late endosomes and/or lysosomes for degradation.

Additionally, the impact on the overall LDLR levels of PCSK9-ΔM2 was compared with PCSK9 LOF variants reported to interfere with PCSK9’s interaction with the R-x-E motif of MHC-I-like proteins, namely PCSK9 R549A-E567A (PCSK9-RE) and R549A-Q554E-E567A (PCSK9-RQE) obtained from the conditioned media of HEK293 cells [13]. As expected, these M2 variants displayed the same LOF phenotype as PCSK9-ΔM2 (Figure 1B), emphasizing the importance of PCSK9’s interaction with an R-x-E motif in “protein X”, e.g., in HLA-C [13]. We next evaluated their effects on the subcellular localization of the LDLR in early endosomes (EEA1 marker). Notably, the incubation of cells with either the control media or those containing LOF PCSK9 derivatives (including PCSK9-ΔM2, PCSK9-RE, and PCSK9-RQE) resulted in an EEA1 punctate signal that was significantly stronger compared to WT PCSK9 (Figure 1B, see red arrows). These findings suggest that different from WT PCSK9, all PCSK9 LOF variants analyzed led to an accumulation of LDLR in early endosomes (see white arrows pointing to merged orange/yellow signals).

The above data demonstrated that PCSK9-ΔM2 variants that do not interact with the putative R-x-E motif in MHC-I-like proteins result in an LOF effect on the extracellular PCSK9-induced LDLR degradation, consistent with recent findings on MHC-I-like molecules in lipid metabolism [13,16]. Accordingly, we suggested that one or more HLA member proteins [17] may serve as “protein X” in regulating PCSK9’s extracellular function. Sequence alignment of MHC-I members showed that six of them (except for HLA-E) contain similarly localized R-x-E motifs in their α1-chain (HLA-A, -B, -C, -F, and -G, and HFE) (Figure 1C). Accordingly, we measured their mRNA expression by qPCR in different hepatocyte cell lines. These data revealed that HLA-A, HLA-B, and HLA-C are the most prevalent members in naïve HepG2 cells (Figure 1D), which are also rich in HepG2-PCSK9 KO cells (Figure 1E) and IHH cells (Figure 1F). In all these cells, HLA-A and HLA-B have higher mRNA expression than HLA-C (Figure 1D,F), as previously reported in other cells of the immune system [27]. The high expression of HLA-A and HLA-B remains in HepG2-HLA-C CRISPR cells lacking endogenous HLA-C (Figure 1G). Concerning HFE, even though its mRNA levels are lower than those of HLA-A, -B, and -C, they are like those of endogenous PCSK9 in naïve HepG2 cells (Figure 1D). Interestingly, our previous RNAseq data in adult mouse liver also revealed that in males and females, PCSK9 and HFE levels are similar. Given the recent findings of the involvement of HFE and HLA-C in lipid metabolism, we have opted to concentrate on these two proteins for further studies.

### 3.2. HFE and HLA-C as New Targets and Regulators of Extracellular PCSK9

To investigate the potential interaction between HFE, HLA-C, and PCSK9, we expressed either HFE (WT or the common HFE-C282Y LOF variant) or HLA-C (WT), along with their common chaperone β2M, into HepG2 PCSK9 KO CRISPR and HepG2 HLA-C KO CRISPR cells, respectively. Subsequently, these cells were incubated with conditioned media of HEK293 cells containing ~300 ng of WT PCSK9 or no PCSK9 (control, empty vector) for 18 h, after which they were collected for Western blot analysis.

Unexpectedly, the data from HepG2 PCSK9 KO cells revealed that in the absence of overexpressed HFE (dark bars), extracellular PCSK9 reduces endogenous LDLR levels by ~50%, but in the presence of HFE/β2M the LDLR reduction is only ~30% (Figure 2A). These data suggested that HFE could act as a negative regulator of the function of extracellular PCSK9 on the LDLR. Interestingly, extracellular PCSK9 also seems to target WT HFE to degradation, as its presence led to a ~60% reduction in its levels (Figure 2A). In contrast, PCSK9 does not significantly enhance the degradation of the HFE-C282Y variant, likely due to its retention in the ER and hence absence from the cell surface [19]. Applying different inhibitors of proteasome degradation (MG132), lysosomal degradation (NH_4_Cl: ammonium chloride), and autophagy (3-MA: 3-methyladenine) suggested that the degradation of HFE occurs in acidic compartments (like the LDLR and HLA-C) (Figure 2B,C). These data revealed that extracellular PCSK9 enhances the degradation of both HFE and LDLR in acidic compartments, but that HFE inhibits the function of PCSK9 on the LDLR.

At the cell surface, HFE usually binds transferrin receptor 1 (TfR1). Under high levels of circulating iron, HFE dissociates from TfR1 to activate the ERK/MAP signaling pathway for hepcidin expression [28,29,30,31]. This dissociation may increase the availability of HFE at the cell surface for its interaction with extracellular PCSK9 and subsequently may result in a higher inhibitory effect on PCSK9’s function on the LDLR. To test this possibility, HepG2 CRISPR PCSK9 KO cells were transfected with WT HFE and then incubated with a PCSK9-enriched medium containing either 200 μg/mL ferric ammonium citrate (FAC) or 200 μM iron chelating factors (DFA: deferoxamine) for 18 h. The addition of DFA dramatically increases the total LDLR levels regardless of the presence of WT PCSK9 due to the presence of an iron regulatory element (IRE) at 3′UTR of LDLR that stabilizes and increases LDLR expression. Intriguingly, in the presence of FAC, PCSK9’s activity on the LDLR is completely blocked by HFE (Figure 2D). Therefore, disrupting the interaction of HFE with TfR1 likely results in higher levels of available HFE at the cell surface and hence greater inhibition of PCSK9’s function on the LDLR.

To further confirm the critical impact of HLA-C on extracellular PCSK9’s ability to enhance the degradation of the LDLR [13], we initially attempted to silence HLA-C expression by siRNA in HepG2 cells. However, we were unsuccessful [19], possibly due to the high endogenous expression levels of HLA-C (Figure 1D). Previously, we found that extracellular PCSK9 did not impact LDLR in CHO-K1 cells, but the reason was unclear. Since CHO-K1 cells were reported to lack endogenous HLA-C expression [32], we compared the effect of extracellular PCSK9 on the LDLR in CHO-K1 cells overexpressing HLA-C-WT-FlagM2 or its **R_68_**-x-**E_70_** mutant HLA-C-R68A-E70A-FlagM2, which no longer binds PCSK9 [13]. Interestingly, HLA-C significantly enhanced PCSK9’s function compared to the control and to the LOF mutant HLA-C-R68A-E70A-FlagM2. In the presence of overexpressed HLA-C in CHO-K1 cells, exogenous PCSK9 significantly reduced LDLR levels by ~40%, supporting HLA-C’s ability to enhance the function of PCSK9. Since these ovarian cells are distinct from hepatocytes, and because of our inability to significantly reduce HLA-C levels from HepG2 cells by siRNA, we completely silenced HLA-C mRNA expression in HepG2 cells using CRISPR-Cas9 technology, thereby generating HepG2 CRISPR HLA-C KO cells (Figure 1G). In the latter, our data confirmed that, in the absence of HLA-C, extracellular PCSK9 has no significant effect on total LDLR levels (either endogenous or overexpressed) (Figure 2E,F). In contrast, an overexpression of HLA-C in these cells recaptured PCSK9’s function towards LDLR degradation and significantly reduced endogenous LDLR levels by ~50% (Figure 2E). Similarly, in these cells, overexpression of HLA-C and LDLR revealed that extracellular PCSK9 also reduced the levels of overexpressed LDLR by ~30% (Figure 2F). These results support the notion that HLA-C is a potential “protein X” candidate implicated in the sorting of the PCSK9-LDLR complex to lysosomes for degradation [13]. Overall, our data further revealed that although HLA-C is critical for PCSK9-induced degradation of the LDLR, it is dispensable for the observed ~30% PCSK9-induced HFE degradation (Figure 2E).

### 3.3. Interaction of PCSK9 with HFE and HLA-C

Recently, the 3D structure of HLA-C’s and PCSK9’s interaction was modeled and confirmed the importance of the **R_68_**-G-**E_70_** motif on HLA-C for the interaction with the M2 domain of PCSK9 [13]. This model predicted that R_68_ and E_70_ on HLA-C bind to E_567_ and R_549_ on PCSK9, respectively [13]. HFE’s structural similarity to HLA-C, along with its similar **R_67_**-V-**E_69_** motif, led us to hypothesize that it could also bind the M2 domain of PCSK9 via its R-x-E motif. Hence, we modeled the M2 domain interaction with HFE using the GRAMM-X web server. The resulting structure proposes that PCSK9 uses a similar R-x-E motif to HLA-C to interact with HFE (Figure 3A). For 5 out of the 10 best models obtained, the M2 of PCSK9’s CHRD is implicated in contacts with HFE, and in 2 of them it is clearly in competition with HLA-C. Of these two models, one model exhibits similar contacts compared to HLA-C. Specifically, R_67_ and E_69_ on HFE are predicted to interact with E_567_ and R_549_ on PCSK9 (Figure 3A). Interestingly, we noticed that the **R_67_**-V-**E_69_** motif in HFE is preceded by a reverse motif **E_64_**-S-**R_66_**, giving the palindromic motif E_64_-x-R_66_-**R_67_**-x-**E**_69_ (Figure 1C). In this model, likely key contacts predicted include R_549_ in the M2 domain of PCSK9 with E_69_ in the α1 domain of HFE, E_567_ in PCSK9’s M2 domain with R_67_ and/or R_78_ in the HFE α1 domain, and Q_554_ in PCSK9’s M2 domain with R_71_ in the α1 domain of HFE (Figure 3A,B). The interactions of HFE with natural PCSK9 variants including Q554E (LOF for LDLR and HLA-C) and H553R (GOF for LDLR and HLA-C) were also modeled (Figure 3B) [13,15]. Such 3D structure modeling suggests that due to the positive charge of R_71_ on HFE, PCSK9 Q554E could enhance HFE’s function, while PCSK9 H553R may hinder it (Figure 3B).

To support the importance of the predicted sites implicated in the HFE-PCSK9 interaction, we mutated/deleted WT PCSK9 (R549A-E567A, R549A-Q554E-E567A, H553R, and ΔM2) and WT HFE (R67A-E69A) to remove the modeled binding sites mentioned above. As predicted, PCSK9 variants lacking the HFE interaction sites were all functionally impaired (LOF) and were not able to degrade HFE (Figure 3C). Interestingly, our data also showed that PCSK9 WT can still interact with HFE R67A-E69A, although it lacks the R-x-E motif. We postulate that this could be because of the presence of the palindromic sequence of E_64_-x-R_66_-**R_67_**-x-**E**_69_ on HFE. Alternatively, our refined 3D structure analysis further suggested the presence of another Arg on HFE (R_78_) that may be closer to PCSK9 E_567_ compared to R_67_ (Figure 3D).

Furthermore, in HepG2 PCSK9 KO CRISPR cells, overexpression of WT HLA-C or its inactive R68A-E70A mutant did not significantly enhance the effect on WT PCSK9 compared to the control (Figure 3E). We presume that the reason for this observation is that the endogenous WT HLA-C is highly expressed in HepG2 cells (Figure 1E) and already reached its maximum effect on PCSK9. Thus, with a low transfection efficiency of HepG2 cells (~20–30%) it would be hard to see an additional or inhibitory effect of overexpressed WT HLA-C or its LOF mutant on endogenous LDLR. Interestingly, HLA-C increased the activity of the extracellular PCSK9 H553R GOF variant on LDLR (Figure 3E). This may suggest that the overexpressed *HLA-C* isoform used (a highly polymorphic gene) may be different than that of the endogenous HLA-C in HepG2 cells and that it may interact better with PCSK9 H553R than WT PCSK9.

As HLA-C and HFE interact with the same residues of PCSK9, we hypothesize that these proteins might be potential competitors. To test this and to eliminate the dominant expression of endogenous HLA-C, we co-expressed HFE and HLA-C in HepG2 HLA-C KO cells. Surprisingly, a similar activity of PCSK9 on both HFE and HLA-C was observed in the absence or presence of the other homologue (Figure 3F,G). These data suggest possible distinct trafficking pathways for HFE compared to HLA-C, and that their localization might be different, and subsequently they may meet PCSK9 in different places/compartments. The superposition of the two models of the HFE/PCSK9 and HLA-C/PCSK9 complexes confirmed that HFE and HLA-C could interact with a similar motif of PCSK9 (R_549_ and E_567_ in M2 domain) via their respective R-x-E motifs (HFE: R_67_ (or R_78_) and E_69_; HLA-C: R_68_ and E_70_) (Figure 3H). Therefore, HFE and HLA-C could potentially interact with PCSK9 depending on their availability, and subsequently either activate (Figure 2E,F) or inhibit (Figure 2A,D) the activity of PCSK9 on the LDLR.

### 3.4. HFE and HLA-C Trafficking

Prior studies indicated that LDLR internalization occurs via clathrin-coated vesicles [10,13,33]. Given HLA-C’s involvement in LDLR degradation by PCSK9, we suggest that HLA-C also uses clathrin-coated vesicles for internalization. Additionally, HFE coupled with TfR1 is also known to undergo internalization through clathrin-coated vesicles [29]. However, an analysis of HFE’s transmembrane domain reveals three unique aromatic Phe residues separated by four residues (F_316_xxxxF_321_xxxxF_326_) that might interact with caveolae [34] (Figure 4I). To determine which endocytosis pathway is dominant for each protein, we knocked down either caveolin 1 (siCav1) or clathrin heavy chain (siCHC) [13] and tested the effect of their absence on the degradation of HLA-C and HFE by PCSK9. As a result, while HLA-C degradation was inhibited by silencing of CHC only, HFE degradation was affected by knocking down either CHC or Cav1 (Figure 4A–C). Since the presence of HFE seems to inhibit PCSK9’s function on the LDLR (Figure 2A,D), we postulated that HFE might compete with the LDLR to interact with PCSK9. To test this hypothesis, we estimated the binding affinity of the LDLR to PCSK9 in the presence or absence of HFE using the CircuLex human PCSK9 functional assay kit (MBL, #CY8153). The data showed that PCSK9 has the same binding affinity to the EGF-A domain of the LDLR either in the presence or absence of HFE/β2M (Figure 4D). Indeed, the absence of the LDLR (using an siRNA approach) inhibits PCSK9 function on HFE (Figure 4E). Therefore, HFE not only fails to compete with the LDLR for interaction with PCSK9 but also depends on the LDLR for its degradation by PCSK9, probably because of the short cytosolic tail of HFE (Figure 4I). This is consistent with prior research indicating that HFE alone cannot internalize into cells and needs to be coupled with another transmembrane protein such as TfR1 [35]. In contrast to HFE and in agreement with a previous study [16], HLA-C degradation by PCSK9 does not require the LDLR (Figure 4F). Indeed, co-expression of LDLR-ΔCT-V5 and HLA-C-FlagM2 in HepG2 HLA-C KO cells showed that in the presence of HLA-C, PCSK9 is better internalized and still active on the LDLR (Figure 4G; compare lanes 6 and 4). However, the absence of the LDLR’s cytosolic tail did not affect the activity of PCSK9 on either HFE or HLA-C (Figure 4H). We suggest that this observation could be due to the presence of endogenous LDLR in those cell lines that could help for the internalization of HFE. This result agrees with the notion that the cytosolic tail of the LDLR is dispensable for the enhanced degradation of the LDLR by PCSK9 [25,26], and further emphasizes the crucial role of “protein X”, e.g., HLA-C, for the sorting of the PCSK9-LDLR complex to degradation compartments.

### 3.5. Modeling of the Interaction between PCSK9’s N-Terminus and HLA-C

Superimposition of the models of the PCSK9/HLA-C and PCSK9/HFE complexes (Figure 3H) shows that HLA-C and HFE both bind the CHRD of PCSK9 via R_549_ and E_567_; however, the orientations of the α1-chain of both HLA-C and HFE in the two complexes are slightly different. The N-terminal end of the PCSK9 prodomain appears closer to the α1-chain of HLA-C than that of HFE (Figure 3H). The possible interaction of PCSK9’s N-terminal peptide with the α1-chain of each HLA member was tested next. This prediction suggested that among all members, only HLA-C’s α1-chain could optimally interact with the N-terminal acidic peptide (aa 31–59) of PCSK9 (Figure 5A). To further support the possible interaction of the above acidic peptide with the antigenic pocket of HLA-C, we evaluated whether this unstructured part of PCSK9 (unstructured in PDB:2P4E) presented a length compatible to reach and possibly bind the antigenic pocket of HLA-C. Thus, modeling of the interaction of the PCSK9 residues 31–59 and the antigenic pocket of HLA-C was carried out. All the 15 models generated by Alphafold suggested an interaction between PCSK9’s unstructured N-terminal residues 38–44 (peptide: YEELVLA) and HLA-C’s peptide-binding pocket (Figure 5B). The mean pLDDT value (used to estimate prediction confidence) for the hotspot interaction region of the best ranked model (residues 37–43, DYEELVL) was 85, suggesting it was modeled with high confidence (Figure 5B). The positioning of PCSK9’s N-terminal structured peptide in an α-helix (PBD:6MV5, complexed with anti PCSK9 fab [36]) was also positioned in this antigenic pocket of HLA-C’s α-1 domain, using GRAMM. This allowed us to evaluate the possible influence of the structure of the acidic peptide on its binding to the HLA-C pocket (Figure 5B). The positioning of the PCSK9 prodomain peptide in the HLA-C pocket suggests possible contacts between Lys_90_ and Arg_93_ of HLA-C with, respectively, Asp_37_ and Glu_40_ for the unstructured peptide and Asp_37_ and Glu_39_ for the structured peptide.

The full PCSK9/HLA-C α1-chain/β2-microglobulin ternary model argues in favor of a plausible interaction of PCSK9/HLA-C simultaneously through PCSK9’s CHRD and N-terminal peptide. The former implicates residues E_567_ and E_549_ of CHRD’s M2 module with residues R_68_ and E_70_ of HLA-C’s α1-chain. The CHRD/α1-chain of HLA-C interaction is also reinforced by the proximity between CHRD’s three M2 domain histidines (537, 553, and 551) and Glu_79_ of HLA-C (Appendix A), which may be enhanced in acidic pHs of endosomes whereupon these His in the M2 domain would be positively charged.

All MHC-I molecules have a similar structure (Appendix A). The presence of positively or negatively charged residues at the HLA-C antigenic pocket (Appendix A) and the electrostatic potential (Appendix A), were evaluated and compared to other MHC-I antigenic pockets. The prevalence of positively charged residues in the HLA-C antigenic pocket supports a possible binding of an acidic peptide such as that of the PCSK9 prodomain.

## 4. Discussion

A detailed understanding of the trafficking and regulation of PCSK9 and LDLR in the liver and other tissues is still lacking. Since the inhibition of PCSK9 presents a potent strategy for treating cardiovascular disorders (CVDs), understanding the detailed trafficking of this protein and its possible implications in other cellular processes holds the potential to extend the advantages of this established treatment beyond CVD [3]. The discovery of HLA-A2 as a new target of PCSK9 led to the combination of PCSK9 inhibitors [16] or antibodies [37] with PD-1 antibodies in cancer therapy. This combination has shown potential in enhancing responses in breast and colorectal cancers compared to PD-1 antibody treatment alone [16,37].

Recently, we introduced a novel model for the clathrin-coated sorting of the PCSK9-LDLR complex that requires at least two partner proteins including CAP1 and an unidentified “protein X”. CAP1 interacts with PCSK9’s M1 and M3 domains, as well as acidic residues in the N-terminal segment of PCSK9’s prodomain [13], thereby exposing the M2 domain of PCSK9 for efficient interaction with “protein X”. In this work, HLA-C is proposed as a bona fide “protein X” candidate for PCSK9’s function on the LDLR [13,14]. Immunofluorescence microscopy revealed that a lack of the M2 domain (more specifically residues R_549_, Q_554_, and E_567_) leading to a loss of the “protein X” interaction with PCSK9 [13], results in a complete LOF of PCSK9 on LDLR degradation, but has no effect on the endocytosis of the PCSK9-LDLR complex (Figure 1A,B). This suggests that “protein X” becomes critical following endocytosis of this complex, likely to sort it to lysosomes for degradation. Additionally, we showed that the presence or absence of these residues is critical for PCSK9’s binding to either HFE or HLA-C (Figure 3A–H). In addition, our cell-based assays in CHO-K1 and HepG2 CRISPR HLA-C KO cells confirmed the crucial role of HLA-C for PCSK9’s function, since in the absence of this protein PCSK9 no longer reduces LDLR levels (Figure 2E,F). Notably, HLA-C still enhances the internalization of the PCSK9-LDLR complex in the absence of the LDLR’s C-terminal cytosolic domain (Figure 4G,I). We hypothesized that this may be due to the presence of a di-Leu motif (Leu-Ile_362_; Figure 4I) reported to be critical for the lysosomal sorting of HLA-C [38]. Previous studies have demonstrated the importance of specific residues at the cytosolic tail of HLA-C, such as serine at position 360 and isoleucine at position 362, for its targeting to lysosomes for degradation [38]. Additionally, our preliminary data in HepG2 CRISPR HLA-C KO cells revealed that both Leu_361_ and the unique Cys_345_ (Appendix A) are needed for HLA-C activation of extracellular PCSK9 function on the LDLR. These data point to the uniqueness of HLA-C in acting as “protein X” via cytosolic tail sequences regulating lysosomal targeting (Leu-Ile_362_) and membrane association (possibly palmitoylation of Cys_345_). Notably, HLA-C also significantly increased the activity of the supposedly GOF PCSK9 H553R variant [13,15] on the LDLR (Figure 3E), supporting the proposed model of the PCSK9-HLA-C interaction where PCSK9’s Arg_553_ interacts better than the native His_553_ with a negatively charged cluster consisting of Glu_79_, Glu_197_, and Glu_201_ in HLA-C (Figure 3B) [13]. To further validate the hypothesis of the importance of the cytosolic tail of HLA-C for PCSK9 function, future studies could use immunofluorescence assays to examine LDLR localization in early endosomes and lysosomes in the presence of cytosolic tail variants of HLA-C. If the HLA-C variants L361A and C345A do not affect the internalization of the PCSK9-LDLR complex but prevent its targeting to lysosomes, as was observed with PCSK9-ΔM2 (Figure 1A), this would further support the role of HLA-C as “protein X” and emphasize the critical roles of cytosolic Cys_345_ and the motif SLI_362_ for targeting the HLA-C-PCSK9-LDLR complex to lysosomes.

The resemblance of HFE’s crystal structure to HLA-C, along with its prior connection to LDLR regulation, motivated us to study its potential regulatory effect on PCSK9’s function. Our 3D modeling and cellular analysis revealed that PCSK9’s R-x-E motif could interact with HFE’s Arg_67_ (or Arg_78_) and Glu_69_, like HLA-C interactions. However, the modeling of PCSK9’s natural variants (Q554E and H553R) suggested that they may exert opposite effects on HFE compared to HLA-C, likely due to the positive charge of Arg_71_ in HFE (Figure 3A,B). Furthermore, our data uncovered a negative regulatory effect of HFE on PCSK9’s function on the LDLR, which could be stimulated under certain physiological conditions such as elevated iron levels (Figure 2A,D). TfR1 binds HFE’s α1 and α2 domains [39]. Our 3D modeling also suggests the involvement of the α1 domain (Arg_67/78_, Glu_69_, and Arg_71_) of HFE in its interaction with PCSK9 (Figure 3A), suggesting a potential competition of TfR1 with PCSK9 to interact with HFE. Elevated iron levels lead to the dissociation of HFE from TfR1, increasing its potential availability for PCSK9 at the cell surface. Apart from the regulatory effect of HFE on extracellular PCSK9, we discovered that this protein is a new target for extracellular PCSK9 for lysosomal degradation, requiring LDLR (Figure 2A–C and Figure 4E), suggesting the possible implication of PCSK9 in iron metabolism.

While HLA-C positively regulates the extracellular activity of PCSK9 on the LDLR, PCSK9 does not rely on HLA-C for HFE degradation, suggesting distinct regulatory pathways of PCSK9 by HFE compared to HLA-C. Previous studies have established that PCSK9 and LDLR internalization occurs through clathrin-coated vesicles [10,13,33]. The present investigation revealed that in HepG2 cells, distinct endocytosis pathways of HFE and HLA-C exist. While the degradation of HLA-C was reduced by the removal of the clathrin heavy chain (CHC), the degradation of HFE was decreased by the absence of either CHC or caveolin (Cav1), suggesting that PCSK9-HFE sorts to lysosomes via clathrin-coated and caveolin-dependent vesicles (Figure 4A–C). The presence of aromatic residues, such as Phe, in the transmembrane domain of HFE supports its potential interaction with caveolae [34].

Accordingly, we propose two internalization pathways for the PCSK9-LDLR complex depending on its interaction with either HLA-C or HFE. Under normal conditions, due to the high expression levels of HLA-C, it interacts with the PCSK9-LDLR complex and sorts it to lysosomes for degradation via clathrin-coated vesicles. However, under elevated iron levels, HFEs may bind the PCSK9-LDLR, and the complex may then be internalized into caveolin-positive endosomes. Such a competitive pathway may prevent the binding of HLA-C to PCSK9 and hence the degradation of the LDLR (Figure 6). It is still unclear why, different from HFE, the caveolin-dependent pathway requires the LDLR for degradation, but the latter is not degraded. Further investigations are needed to elucidate how, in the presence of HFE, the LDLR can recycle back to the cell surface, and how solely HFE undergoes degradation. One approach could involve conducting immunofluorescence assays to examine the localization of the LDLR, either with lysosomal hydrolases (e.g., cathepsins) or intracellular markers involved in LDLR recycling pathways (e.g., Rab11 for slow recycling and Rab4 for fast recycling). Regarding HFE degradation, assessing the significance of the VL motif at the cytosolic tail of this protein [40] could provide insights into whether this di-aliphatic motif is critical for HFE trafficking to acidic compartments for degradation, or if it relies on another yet-unidentified protein for lysosomal sorting.

Similar pathways have been observed for the TGF-β receptor, in which its internalization can occur either through clathrin-coated pits (for signal transduction) or caveolin-positive vesicles (for its degradation), where some physiological conditions, like high potassium levels, could favor one pathway over the other [41]. Future work could focus on identifying the factors that dictate the selection of either pathway for PCSK9 function. For instance, investigating the internalization of HFE via the caveolin-dependent pathway could involve mutating potential residues involved in caveolar interaction, introducing iron into our experimental conditions, or removing TfR1 protein to prevent HFE internalization via clathrin-coated vesicles. Understanding how these manipulations affect HFE trafficking and subsequently impact PCSK9 activity and LDLR levels would provide valuable insights to better predict PCSK9’s function under various physiological conditions.

Our results and molecular models advocate in favor of a more general crosstalk between PCSK9 and HLA molecules, which may have profound implications in immunology. Our current model for the PCSK9/HLA-C complex proposes that PCSK9’s acidic N-terminus could bind to the peptide-binding pocket of the latter (Figure 5A,B). Future works using site-directed mutagenesis on the proposed interface should provide more substantial evidence for the validation or refutation of this model and the possible regulation of the levels of HLA-C at the cell surface by the acidic domain of PCSK9. Additionally, the possible implication of the acidic domain of PCSK9 in the regulation of its intracellular activity and its effect on antigen presentation by HLA members and/or their cell surface localization call for a more detailed analysis of these phenomena.

While the intracellular activity of PCSK9 remains poorly understood, several studies suggested its distinctiveness from its extracellular function (Appendix A). The presence of the M2 domain of PCSK9 appears to be non-critical for its intracellular activity [12]. Consequently, HFE or HLA-C may not exert the same effect on the intracellular activity of PCSK9. Our unpublished data indicate that HFE has no significant impact on the intracellular activity of WT PCSK9 on the LDLR. However, the overexpression of HFE markedly increased PCSK9 levels both in lysate and media. Notably, unlike extracellular PCSK9, overexpressed PCSK9 fails to degrade HFE. The proteins regulating the intracellular function of PCSK9 still need to be identified.

## 5. Conclusions

In summary, our findings highlight the significance of PCSK9’s M2 domain and HLA-C in the extracellular activity of PCSK9 for lysosomal degradation of LDLR. Furthermore, our results suggest an opposite regulatory effect of HFE compared to HLA-C, which utilizes different endocytosis pathways when interacting with PCSK9. The interaction of each protein could potentially influence the fate of PCSK9-bound proteins in the cellular trafficking process.

## Figures and Tables

**Figure 1 cells-13-00857-f001:**
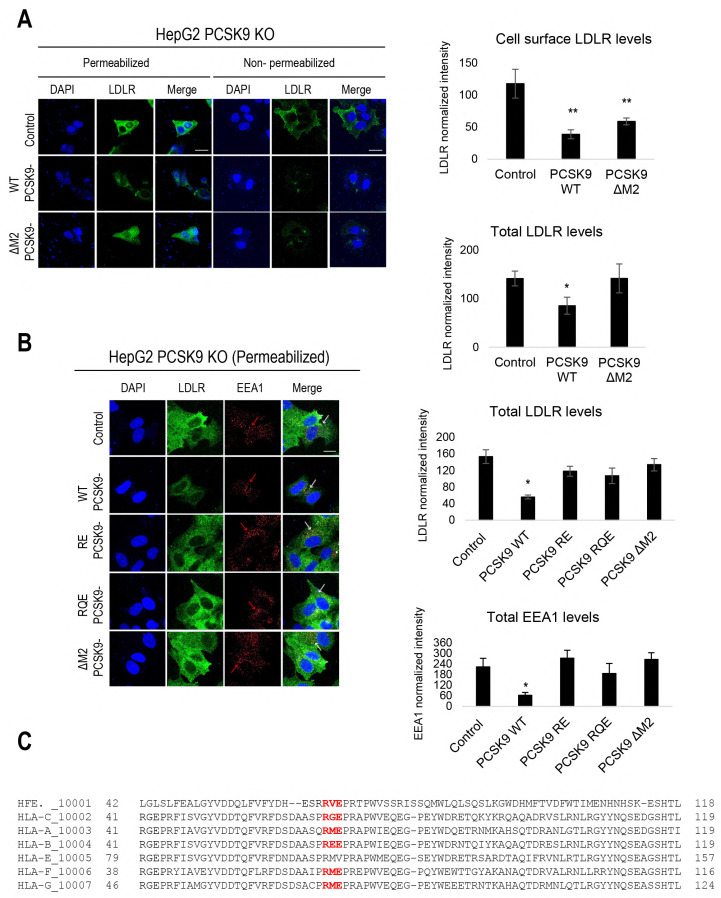
Extracellular importance of M2 domain of PCSK9 and mRNA expression of HLA members: (**A**) Immunofluorescence staining of total LDLR (permeabilized conditions) vs. cell surface LDLR (non-permeabilized conditions) levels in the presence of PCSK9-ΔM2 compared to WT PCSK9. (**B**) Immunostaining of total LDLR along with early endosomal (EEA1) marker in the presence of PCSK9 ΔM2, PCSK9 R549A-E567A (RE), and PCSK9 R549A-Q554E-E567A (RQE). Blue: nuclei; green: LDLR; red: EEA1. Scale bar: 20 μm. Data are representative of three independent experiments. Quantifications are averages ± standard deviation (SD). * *p* < 0.05; ** *p* < 0.01 (two-sided *t*-test). In each separate experiment, approximately 10 pictures per condition were captured using confocal microscopy. Within each picture, approximately 3–15 cells were analyzed (the mean intensity values were measured) for quantification. (**C**) Location of R-x-E motif in the amino acid sequence of each HLA member. mRNA levels of HLA protein and PCSK9 in (**D**) HepG2-naïve, (**E**) HepG2 CRISPR PCSK9 KO, (**F**) IHH, and (**G**) HepG2 CRISPR HLA-C KO cells by qPCR experiment. mRNA values were normalized to the mRNA expression of the housekeeping gene (TATA box binding protein: TBP) to calculate the relative mRNA expression of each MHC-I member and compare it to the expression of PCSK9 and β2M.

**Figure 2 cells-13-00857-f002:**
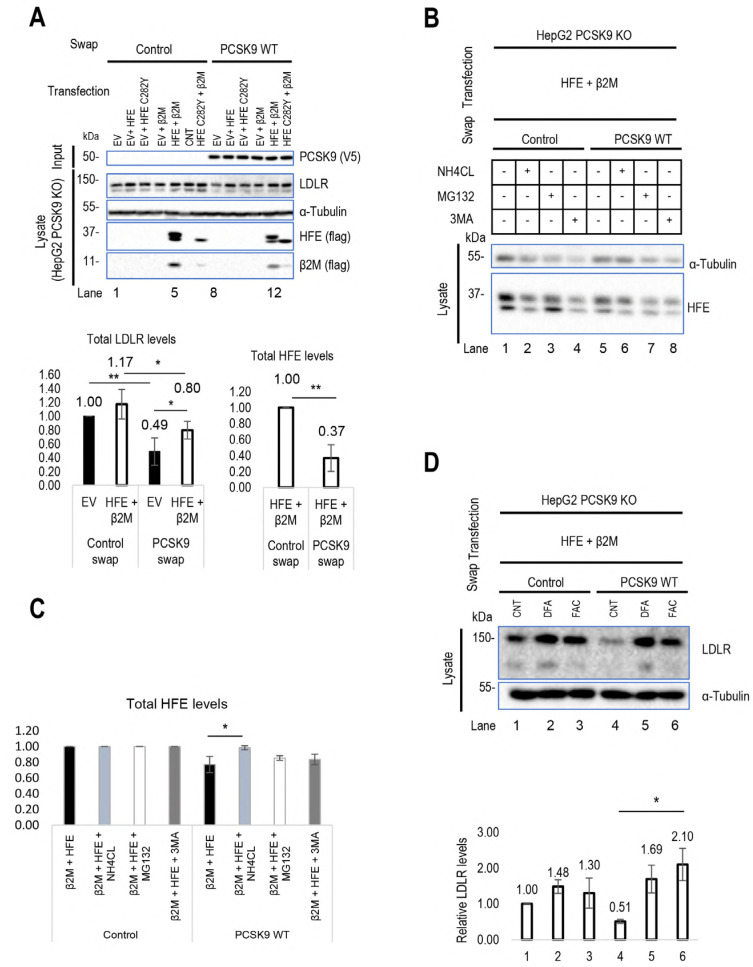
Regulatory effect of HLA-C and HFE on PCSK9 and vice versa: (**A**) The effect of cell surface HFE on extracellular PCSK9. HepG2 lacking endogenous PCSK9 was transiently transfected with an empty vector (EV), HFE-WT-flagM2, HFE-C282Y-flagM2, and β2M-flagM2 and incubated with conditioned media enriched with extracellular PCSK9 or empty vector (control). The effect of HFE on the extracellular activity of PCSK9 has been analyzed by WB (SDS/PAGE on 8% tris-glycine gel) analysis. The quantification of total LDLR and HFE levels is shown in respective charts. (**B**,**C**) Cellular inhibitors were used to inhibit lysosomal degradation (NH4CL: ammonium chloride), proteasome degradation (MG132), or autophagy (3-MA: 3-methyladenine). The data suggest the possible lysosomal degradation of HFE. (**D**) Effect of iron on HFE function. HepG2 PCSK9 KO cells were transfected with WT HFE and β2M, then incubated with conditioned media from HEK293 cells expressing an empty vector (control) or WT PCSK9. Following the incubation with conditioned media, cells were treated with either ferric ammonium citrate (FAC) or deferoxamine (DFA) to analyze the function of HFE on PCSK9 in different iron conditions. (**E**) HLA-C’s impact on extracellular PCSK9 in HepG2 HLA-C KO cells. These cells were transfected with an empty vector (EV), WT HLA-C, or WT HFE and then incubated with conditioned media from HEK293 cells expressing an empty vector (control) or WT PCSK9. The total levels of endogenous LDLR were quantified. (**F**) HepG2 HLA-C KO cells were co-transfected with (empty vector (EV) + WT LDLR), (WT HLA-C + WT LDLR), or (WT HFE + WT LDLR) and then incubated with conditioned media from HEK293 cells expressing an empty vector (control) or WT PCSK9. The total levels of overexpressed LDLR were quantified. All cell lysates were extracted to be analyzed by WB (SDS/PAGE on 8% or 6.5% tris-glycine gel). Data are representative of three independent experiments. Protein levels were normalized to the control protein, α-tubulin. Quantifications are averages ± standard deviation (SD). * *p* < 0.05; ** *p* < 0.01 (two-sided *t*-test).

**Figure 3 cells-13-00857-f003:**
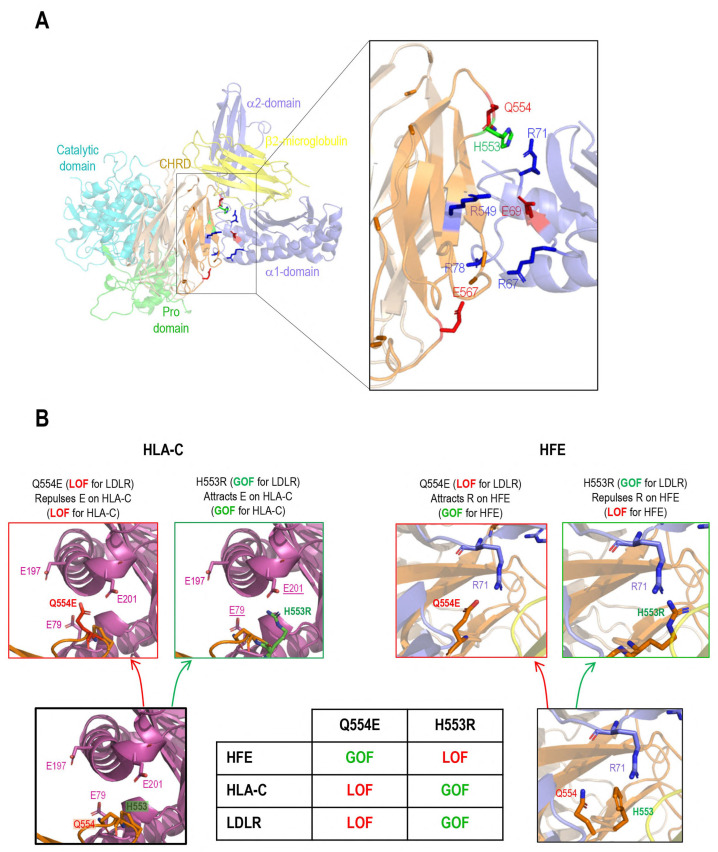
PCSK9’s direct interaction with HLA-C compared to HFE: (**A**) Molecular modeling of the interaction of the M2 subdomain of PCSK9 with the α1 domain of HFE suggests the interaction of the R_549_-x-E_567_ motif of PCSK9 with the R_67_-x-E_69_ motif of HFE. (**B**) Further analysis of the 3D modeling of PCSK9’s interaction with HFE (like what has been published before for HLA-C [13]) suggests the presence of another interaction site on HFE (R_71_) with PCSK9 that could be sensitive to PCSK9’s natural mutations Q554E (GOF for HFE) and H553R (LOF for HFE). The model of HLA-C’s interaction with Q554E and H553R is adopted from [13]. (**C**) HepG2 PCSK9 KO cells were transfected with an empty vector (EV), WT HFE, or the HFE R67A-E69A variant and then incubated with conditioned media from HEK293 cells expressing an empty vector (control), WT PCSK9, and proposed LOF variants on PCKS9 (ΔM2, R549A-E567A, R549A-Q554E-E567A, and H553R). Cell lysates were extracted to be analyzed by WB (SDS/PAGE on 8% tris-glycine gel). The cell-based data confirmed the interaction sites predicted by 3D structure modeling. (**D**) Further 3D structure modeling revealed a second arginine at position 78 that could interact better with PCSK9 E_567_. (**E**) HLA-C’s interaction with PCSK9. HepG2 PCSK9 KO cells were transfected with an empty vector (EV), WT HLA-C, or the HLA-C R68A-E70A variant and then incubated with conditioned media from HEK293 cells expressing an empty vector (control), WT PCSK9, and proposed LOF variants on PCKS9 (ΔM2, R549A-E567A, R549A-Q554E-E567A, and H553R). Cell lysates were extracted to be analyzed by WB (SDS/PAGE on 8% tris-glycine gel). The cell-based data confirmed the interaction sites predicted by 3D structure modeling [13]. (**F**,**G**) Studying the potential competition of HFE with HLA-C to interact with PCSK9 and get degraded. HepG2 HLA-C KO cells were co-transfected with empty vector (EV), WT HLA-C, WT HFE, or WT HLA-C + WT HFE and then incubated with conditioned media from HEK293 cells expressing an empty vector (control) or WT PCSK9. Total levels of HFE and HLA-C were quantified in respective charts. (**H**) Three-dimensional modeling of possible similarities of HFE with HLA-C and their interaction with PCSK9. All protein levels were normalized to the control protein, α-tubulin. Data are representative of three independent experiments. Quantifications are averages ± standard deviation (SD). * *p* < 0.05; ** *p* < 0.01 (two-sided *t*-test).

**Figure 4 cells-13-00857-f004:**
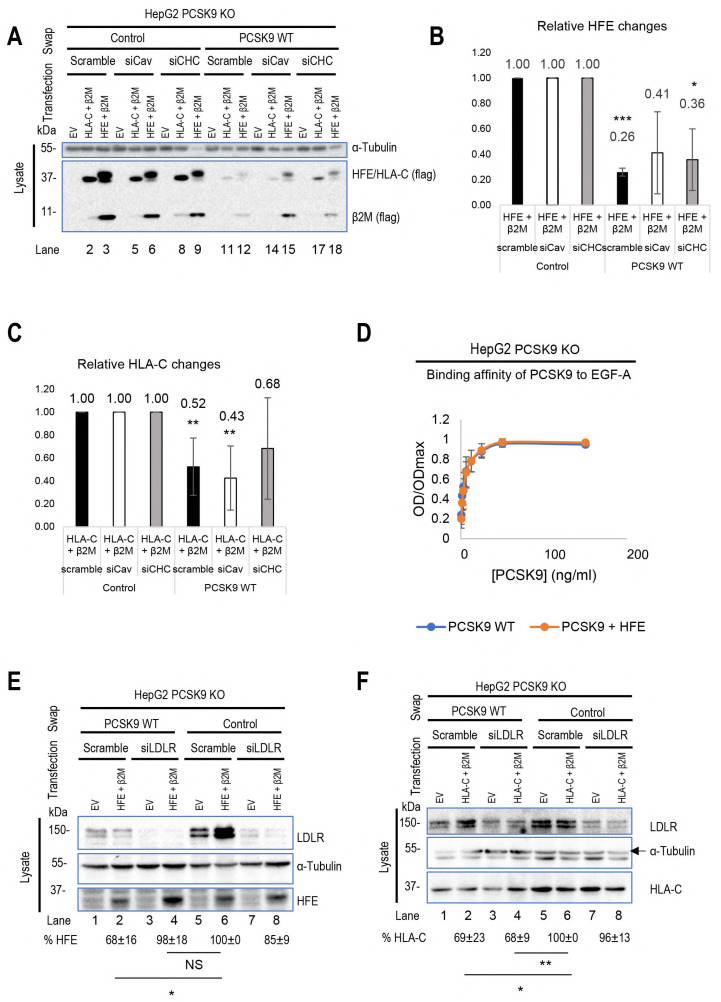
Distinct trafficking of HFE compared to HLA-C: (**A**–**C**) HepG2 HLA-C KO cells were transfected with siRNA against clathrin heavy chain (siCHC), siRNA against caveolin 1 (siCav1), or non-targeting siRNA. After 24 h, these cells were transfected with an empty vector (EV), WT HFE, or WT HLA-C and then incubated with conditioned media from HEK293 cells expressing an empty vector (control) or WT PCSK9. (**D**) The binding affinity of WT PCSK9 to the LDLR was measured using the CircuLex human PCSK9 functional assay kit. The results of the affinity curve suggest that the presence of HFE/β2M does not affect the interaction of PCSK9 with the EGF-AB domain of LDLR. (**E**,**F**) HepG2 PCSK9 KO cells were transfected with siRNA against LDLR or non-targeting siRNA (Scramble). After 24 h, these cells were transfected with either an empty vector (EV) or WT HLA-C/HFE and then incubated with conditioned media from HEK293 cells expressing an empty vector (control) or WT PCSK9. The effect of LDLR on either HFE or HLA-C was measured. (**G**,**H**) HepG2 HLA-C KO cells were co-transfected with (empty vector (EV)+ ΔCT LDLR), (WT HLA-C+ ΔCT LDLR), or (WT HFE + ΔCT LDLR) and then incubated with conditioned media from HEK293 cells expressing an empty vector (control) or WT PCSK9. The effect of ΔCT LDLR on either (**G**) PCSK9 internalization or (**H**) HFE/HLA-C degradation was measured in prospective charts. (**I**) Comparison of cytosolic and transmembrane domains of HFE with other major HLA family members. Notice the unique residues (in red) in the transmembrane domain of HFE and in the cytosolic tail of HLA-C (in red surrounded by green boxes). All cell lysates were extracted to be analyzed by WB (SDS/PAGE on 8% tris-glycine gel). All protein levels were normalized to the control protein, α-tubulin. Data are representative of three independent experiments. Quantifications are averages ± standard deviation (SD). * *p* < 0.05; ** *p* < 0.01; *** *p* < 0.001 (two-sided *t*-test).

**Figure 5 cells-13-00857-f005:**
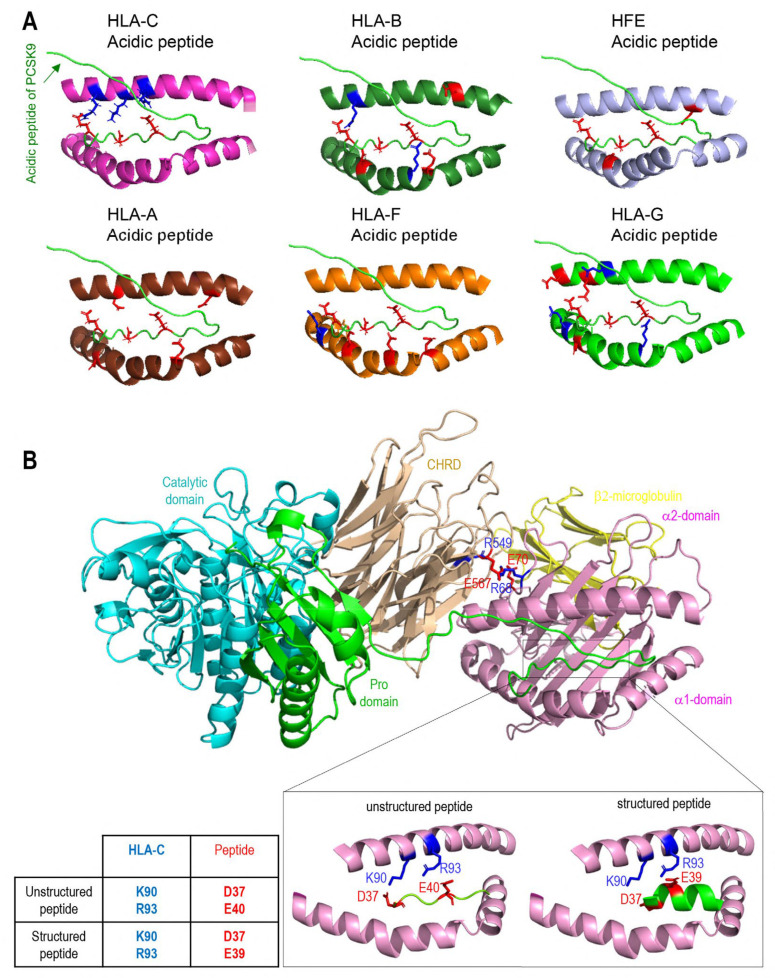
Modeling of interaction between PCSK9’s N-terminus with HLA members: (**A**) Molecular modeling of the interaction of PCSK9’s 31–59 propeptides with all HLA proteins. (**B**) Molecular modeling of the interaction of PCSK9’s 31–59 unstructured propeptides (green) with HLA-C antigenic pocket of HLA-C (magenta). A zoomed view shows possible proximities between basic amino acids of HLA-C pocket (blue) and acid residues of unstructured (left) or structured (right) peptides (red).

**Figure 6 cells-13-00857-f006:**
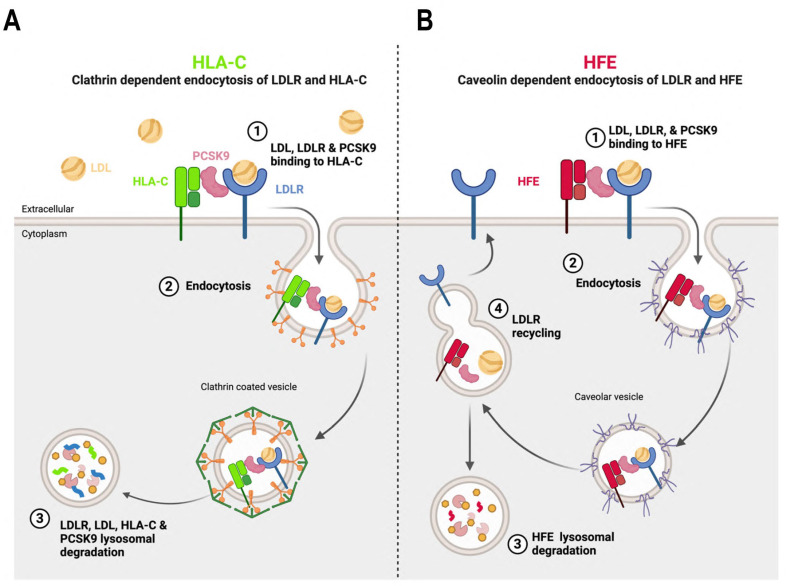
The proposed model of PCSK9’s regulation by either HFE or HLA-C. The model was created by using BioRender.com. (**A**) In normal conditions within hepatocytes, where the expression of HLA-C is significantly higher compared to HFE, it interacts with the M2 domain and prodomain of PCSK9. The resulting complex of PCSK9-LDLR-HLA-C is internalized via clathrin-coated vesicles, which transport the entire complex to lysosomal compartments for degradation. (**B**) However, in certain physiological conditions, such as elevated levels of circulating iron, HFE could also interact with PCSK9. In this scenario, the entire complex is likely internalized with the assistance of caveolin-dependent vesicles. This alternative endocytosis pathway leads LDLR to recycle back to the cell surface, while only HFE is directed towards degradation.

## Data Availability

Data are contained within the article.

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
