# Peer review of "Insights into PCSK9-LDLR Regulation and Trafficking via the Differential Functions of MHC-I Proteins HFE and HLA-C"

_cells, 2024, doi:10.3390/cells13100857_

Round 1

Reviewer 1 Report

Comments and Suggestions for Authors

Doctor Mikaeeli and colleagues present a study about PCSK9-LDLR regulation and trafficking via the 2 differential functions of MHC-I proteins HFE and HLA-C

 Overall, the article is commendably written, featuring numerous tests and results that substantiate the hypothesis.

I have a few questions:

The results section appears extensive while the discussion seems relatively brief. It appears that certain segments of the results might better fit within the discussion section to provide a more cohesive narrative.

Which strategy do the authors plan to employ to demonstrate that the phenomena replicated in cell lines accurately reflects occurrences in individuals?

Do these findings and the proposed two-pathway approach have any implications for individuals currently using PCSK9 inhibitors?

What are the limitations of this study?

What is the primary contribution of the article, and how does it translate into tangible applications in clinical practice?

Author Response

Dear Reviewer,

Please see the attachment,

Thank you for your consideration,

Reviewer 2 Report

Comments and Suggestions for Authors

This manuscript by Mikaeeli and colleagues showed in details the mechanisms of regulation of PCSK9-LDLR complex by MHC-1 proteins HFE (inhibition of the activity of PCSK9 on LDLR) and HLA-C (promotion of the activity of PCSK9 on LDLR). The manuscript would be relevant for a better understanding of the factors regulating the recycling of LDLR and consequently the levels of lipids in the blood, but shows serious limitations in the way data are presented and analyzed.

Main concern:

Statistical analysis is conducted in superficial and unclear way in many cases. At the end of each figure there is always the same text: ``quantifications are averages standard deviation (SD). * p<0.05; ** p<0.01; ***p<0.001 (two-sided t-test). NS: non-significant. For example, in figure 1 what the statistic is referred to?  In figure 4B what treatments are compared?  Most importantly in all the graphs, the authors used to compare condition x to condition y setting condition x as 1 (i.e relative levels of HFE protein analyzed by WB). But condition X does not have any standard deviation. SD should be present also in the control group to make the statistic and observations reliable. Otherwise, authors can use the total OD values detected. In other cases, statistic would be beneficial to make the data more convincing. For examples for figure 1 LDLR staining quantification of a higher number of cells is required plotting the results on graphs.  Same issue with figure 1B where the difference in the EEA1 punctate signals are not so evident (expecially control vs WT-PCSK9). The authors can use some specific method to quantify the level of co-localization of LDLR and EEA1 marker (i.e Pearson correlation coefficient).

Some concerns are present also in few western blotting:

Figure 2b: samples are not equilibrated and differences in tubulin levels reflects the differences in HFE levels.

Figure 2e: the LDLR western do not correspond to any of the provided images of original WB. Moreover, the n3 are very different between each other regarding the levels of LDLR.

Figure 3 panel C. In the n3 used for quantifications mainly no bands are present for LDLR and the order of samples is mixed. Why?

Figure 4 panel G. The levels of LDLR-DCT are shown. But in this case LDLR is presented as single band, while in other WB is presented as a double band. What is the criteria to show one or two bands and the authors quantified the levels of LDLR on one band or both? Overall is very difficult to properly evaluate the manuscript when data are presented and analyzed not in a solid way. 

 Minor points:

-Would be beneficial for the readers, a better presentation of the PCSK9 protein with all its domains depicted in a figure and more detailed info about its intracellular processing (PreproPCSK9/ProPCSK9/mpcsk9 secreted).

-In the western blots of LDLR, and HFE double bands are present. Which one is used for the quantification?

-regarding the incubation of HepG2 PCSK9 KO CRISPR cells with PCSK9-enriched media derived by HEK293 transfected with different vectors. Were the levels of PCSK9 similar in all the supernatants tested by ELISA? If not, how the authors equilibrated the samples?

Comments on the Quality of English Language

Minor editing required

Author Response

(The authors gave the same response as above.)

Round 2

Reviewer 2 Report

Comments and Suggestions for Authors

The authors replied to my comments. I have few considerations to further improve the manuscript:

1) Authors performed quantification of LDLR staining but no information are provided on how the quantification was done. In the graphs what is on the y axis? Mean intensity value? The text report that values are the average and SD of three experiments but counting how many cells for experiments? This should be mentioned in the figure text and materials and methods

2) Regarding the western blotting quantification the limited time availability to revise the manuscript is not a valid reason to not perform further experiments or analysis. What authors wrote after that makes much more sense because, by my own experience, I know how difficult is to get statistically significant data on triplicate of WB, since you can observe an increase of the protein X in condition B compared to A in all 3 westerns but still not get any statistically significant difference because the background and band intensities are different among WB. In materials and methods authors should mention how normalization of WB samples was performed.

3) There are little mistakes to correct through the text. I spotted these:

- In figure 1 legend is written that the scale bar is 1μm but I do not see the scale bar in the pictures. NS as non-  significant does almost never appear in the graph.

- In figure 4D why the x scale bar has negative number?

-

Author Response

Please find our response in the attached .pdf file below
